# Quality Characteristics of Vegan Mayonnaise Produced Using Supercritical Carbon Dioxide-Processed Defatted Soybean Flour

**DOI:** 10.3390/foods13081170

**Published:** 2024-04-11

**Authors:** Kyo-Yeon Lee, Chae-Yeon Han, Wasif Ur Rahman, Nair Chithra Harinarayanan, Chae-Eun Park, Sung-Gil Choi

**Affiliations:** 1Department of Food Science and Technology, Institute of Agriculture and Life Sciences, Gyeongsang National University, Jinju 52828, Republic of Korea; leeyeon0511@naver.com; 2Upland Crop Breeding Research Division, Department of Southern Area Crop Science, National Institute of Crop Science, RDA, Miryang 50424, Republic of Korea; coddus1229@naver.com; 3Division of Applied Life Science (BK21), Gyeongsang National University, Jinju 52828, Republic of Korea; wur1497@gnu.ac.kr (W.U.R.); chithran2023@gmail.com (N.C.H.); chaechae_0621@naver.com (C.-E.P.)

**Keywords:** mayonnaise, soybean, emulsifier, supercritical carbon dioxide, vegan

## Abstract

Emulsifiers, like egg yolk (EY), are necessary for the formation of mayonnaise, which is an oil-in-water type of colloid. This study aimed to assess the potential of defatted soybean powder treated with supercritical carbon dioxide (DSF) to enhance the quality of plant-based mayonnaise as plant-based alternatives gain popularity. This study involved the production of DSF and the comparison of its quality attributes to those of mayonnaise made with varying amounts of control soy flour (CSF), DSF, and EY. It was found that mayonnaise made with an increased quantity of DSF showed better emulsion stability, viscosity, and a smaller, more uniform particle size when compared with CSF mayonnaise. Additionally, DSF mayonnaise was generally rated higher in sensory evaluation. The addition of approximately 2% DSF positively influenced the emulsion and sensory properties of the vegan mayonnaise, indicating that DSF is a promising plant-based alternative emulsifier for the replacement of animal ingredients.

## 1. Introduction

Mayonnaise is one of the oldest and most widely used acidic condiments in the world, and is used to enhance the texture and flavor of food. It is a classic oil-in-water (O/W) colloidal structure, containing 65–80% oil and is typically formed by slowly mixing egg yolk (EY) with oil along with vinegar for pH adjustment and other ingredients such as salt, sugar, and mustard [1].

EY functions as an emulsifier in mayonnaise, containing low-density lipoprotein, high-density lipoprotein, phospholipids, and unbound proteins, which help in forming granular micro-oil particles and in preventing coagulation, thereby imparting the desired texture [2,3]. Although EY is an excellent emulsifier, with strong emulsifying properties, it has disadvantages, such as high cholesterol content, potential contamination with *Salmonella*, and high cost. Furthermore, the global trend toward health, environmental preservation, and animal ethics, along with the growing vegan market, highlights the need for plant-based emulsifiers [4]. Consequently, research is actively underway to explore and develop emulsifiers that can replace EY in mayonnaise production [5,6]. Constant research efforts are being made to develop mayonnaise with plant-based ingredients, such as wheat protein [7], soy milk [8], pea [9], chickpea, faba bean, lentil [10], aquafaba [11], and clover sprouts [12].

Soybeans, comprising 40% protein, 15% monosaccharides, 15% dietary fiber, 20% oil, and 10% other components, serve as a high-quality protein source with low cholesterol and lactose levels, making them suitable as a substitute for animal ingredients [13]. The principal storage proteins in soy, glycinin, and β-conglycinin, account for > 80% of its protein content, with β-conglycinin known to have a remarkable structural stability at oil–water interfaces [14]. However, soybeans have a drawback of high fat content (20%). It has been reported that soy milk made from full-fat soybean flour presents challenges when completely substituting the EY in mayonnaise production [8]. Soybean flour has been indicated to have lower emulsifying qualities than soy protein isolates [15]. Studies have also shown that the removal of fat from mustard, soy, and flaxseed powders increases water and oil absorption capacity, foamability, and emulsifying properties, thereby enhancing the functional characteristics of the powders [16]. Fat removal appears to be one strategy by which to enhance the functional qualities of soybean powder for use as an emulsifying agent.

The primary method for removing fats from soybeans is organic solvent extraction, typically using hexane. Although this method is efficient in oil extraction and solubility, concerns about residual hexane and its high flammability raise environmental and safety issues. Supercritical fluid extraction using carbon dioxide as a solvent is harmless to humans and facilitates the separation of solvent and solute, making it an environmentally friendly and safe alternative to traditional organic solvent extraction [17,18]. Studies have demonstrated that supercritical processing can produce soy protein with low denaturation and superior nutritional, functional, and biodegradable qualities (49–50% protein concentration) [19]. Although defatted soybean powder, processed with supercritical treatment, has been established to have better functional properties, in terms of moisture and oil absorption, emulsification, and foaming, than hexane-treated defatted soybean powder [20], research on the quality characteristics of mayonnaise made with supercritically treated defatted soybean powder is lacking.

The aim of this study was to make vegan mayonnaise using soybean powder. Mayonnaise was prepared using full-fat control soybean flour (CSF), defatted soybean flour (DSF) treated with supercritical fluid, and EY. Their emulsifying and qualitative attributes were examined. This study aimed to assess the potential for the quality enhancement of vegan mayonnaise using supercritical treatment.

## 2. Materials and Methods

### 2.1. Materials and Chemicals

Soy flour (Always Green Co., Icheon, Republic of Korea), soybean oil (Ottogi Co., Seoul, Republic of Korea), EY (Edentown Co., Ltd., Incheon, Republic of Korea), white sugar (CJ, Seoul, Republic of Korea), sea salt (Sinsong Co., Seoul, Republic of Korea), yellow mustard powder (Ottogi Co., Seoul, Republic of Korea), and brewed vinegar (Ottogi Co., Seoul, Republic of Korea) were purchased. All other chemicals used were analytical-grade reagents.

### 2.2. Supercritical Carbon Dioxide (SC-CO_2_) Processing

For the defatting of soy flour, a SC-CO_2_ extraction system (Ilshin Autoclave Co. Ltd., Daejeon, Republic of Korea) was used. The system consists of an extraction vessel, a separation vessel, a high-pressure pump, and a CO_2_ storage tank. An amount of 500 g of soybean flour was added to the extraction vessel and connected to the extractor. The extraction was conducted at 45 °C and 40 MPa for 10 h with a CO_2_ flow rate of 30 g/min, after which the supercritical CO_2_ was separated into gaseous CO_2_ and oil through a micrometer valve in the separation vessel [20]. The defatted soybean flour was kept at −18 °C for experimental use.

### 2.3. Proximate Composition Analysis

The proximate composition of the CSF, DSF, and EY was analyzed according to the AOAC method [21]. Moisture content was ascertained using atmospheric pressure drying at 105 °C, and ash was evaluated by direct ashing at 550 °C. The auto Kjeldahl method was utilized for crude protein, while the Soxhlet extraction method was used for assessing crude fat.

### 2.4. Mayonnaise Preparation

The formulation for mayonnaise samples with 3 different emulsifiers (CSF, DSF, and EY) is presented in Table 1. Traditional mayonnaise, made with egg yolk as an emulsifier, was used as a reference. The mayonnaise samples were prepared as per the methods of Lee et al. [22], with slight modifications. The emulsifiers were used at levels of 0.5, 1, 1.5, 2, and 3%. Initially, all ingredients, excluding the oil and vinegar, were mixed using a direct-driven digital stirrer (PL-SS41D, Poonglim, Republic of Korea) for 1 min at 700 rpm. A coarse emulsion was formed by adding the oil gradually to the aqueous mixture containing sugar, salt, mustard powder, water, and emulsifier. The speed was adjusted to 15,000 rpm with continuous dropwise addition of oil. Ultimately, brewed vinegar was added to the mayonnaise sample and mixed at 700 rpm for 1 min. Final mayonnaise samples were stored at 4 °C for further analysis.

### 2.5. Color Value

The color values of the mayonnaise samples were measured using a colorimeter (Minolta CT-310, Osaka, Japan) calibrated with a standard white plate (L = 93.6, a = 0.31, b = 0.32). Each sample was placed in a Petri dish (60 × 15 mm; SPL Life Science Co., Pochen, Republic of Korea) and measured three or more times, with results provided as L (lightness), a (−: greenness; +: redness), and b (−: blueness; +: yellowness) values.

### 2.6. Confocal Laser Scanning Microscope

The microstructures of the mayonnaise samples were compared using a confocal laser scanning microscope (CLSM; FV1000 MPE, Olympus America, Inc., Melville, NY, USA). Each sample (1 g) was stained with 10 μL of 0.1% Nile red and 10 μL of 0.1% Nile blue. Activation of Nile blue and Nile red occurred at the Ar laser 488 nm and the He–Ne laser 633 nm, respectively.

### 2.7. Particle Size Analysis

The particle size was measured using a laser-based particle size analyzer, Mastersizer 3000 (Cilas Corp., Orleans, France), installed at the Central Research Facilities of Gyeonsang National University using the method of Metri-Ojeda et al. [23], with minor modifications. The mean particle diameters measured were processed using the SizeExert software (ver. 9.51, Cilas Corp., Orleans, France).

### 2.8. Measurement of Emulsion Stability

The emulsifying capacity of the plant-based mayonnaise was determined as per the methods of Lee et al. [22], with slight modifications. Emulsion stability (thermal stability) was assessed by shaking the plant-based mayonnaise in a shaking water bath (Baths WSB shaking water bath) at 185 rpm for 1 h at 90 °C, followed by centrifugation at 3000× *g* for 30 min.
(1)$$\mathrm{Emulsion stability}\;(\%)\;=\;\frac{\mathrm{The volume of the emulsified layer after centrifugation}}{\mathrm{The total volume of the emulsion}}\;\times 100$$


### 2.9. Viscosity Measurement

The viscosity of the plant-based mayonnaise was determined using a Brookfield viscometer (DV II+, Brookfield Engineering Labs Inc., Stoughton, MA, USA). The spindle used was LV4, and the spindle speed was 20 rpm. The experiment was performed three or more times.

### 2.10. Texture Analysis

The texture of the mayonnaise was measured using a Texture analyzer (Stable Micro stable system Ltd., TA-XT Express, Surrey, UK) with a 20 mm diameter cylinder probe in a two-bite compression test. The parameters were as follows: pretest speed 1.0 (mm/s), test speed 1.0 (mm/s), post-test speed 1.0 (mm/s), test distance 3.0 (mm), and trigger force 3.0 (g). The force–distance curve obtained was analyzed using Texture Expert software software version 2.54 for windows. The experiment was performed three or more times.

### 2.11. Sensory Evaluation

Sensory evaluations of the mayonnaise samples were conducted with 20 undergraduate and graduate students from Gyeongsang National University. Participants were given mayonnaise samples, along with plain crackers (IVY, Haitai Co., Ltd., Seoul, Republic of Korea). Evaluation criteria included appearance, aroma, taste, texture, and overall satisfaction, rated on a nine-point hedonic scale (1 = “dislike extremely,” 9 = “like extremely”). Samples were assigned random three-digit numbers and tasted in random order. Water was provided between samples for accurate taste evaluation.

### 2.12. Statistical Analysis

Experimental data are presented as mean ± standard deviation from three or more repeated trials. Data were analyzed using SPSS 12.0 (SPSS, Inc., Chicago, IL, USA) and subjected to analysis of variance with *p* < 0.05 regarded as statistically significant, followed by Duncan’s multiple range test to verify significance.

## 3. Results and Discussion

### 3.1. Proximate Composition Analysis

The moisture, ash, crude protein, and crude fat contents of each emulsifier used in the mayonnaise are listed in Appendix A. The moisture content is a critical factor for the long-term stability of the product. The moisture contents of the emulsifiers were found to be CSF (7.41% ± 0.65%), DSF (3.30% ± 0.08%), and EY (3.48% ± 0.24%), with DSF showing a 55.47% decrease compared with CSF. It is considered safe to store with moisture content below 12% [24]. The moisture contents of flaxseed and hexane-defatted flaxseed were found to be 7.94% ± 0.33% and 9.37% ± 0.40%, respectively, suggesting an increase in moisture with hexane treatment [25]. SC-CO_2_ defatting appears to be a technique that can reduce the moisture content of powders. Ash content, indicating the mineral content per emulsifier, was noted as (5.41% ± 0.29%) for CSF, DSF (6.26% ± 0.14%), and EY (3.94% ± 0.21%).

The crude fat contents of the emulsifiers were 18.95% ± 0.95% for CSF, 0.51% ± 0.37% for DSF, and 54.27% ± 0.56% for EY samples, with DSF showing an approximately 97.31% decrease compared with CSF. The crude fat contents in mustard, flaxseed, and soybean seed meal powders were reported to be 7.26% ± 0.16%, 4.77% ± 0.35%, and 1.20% ± 0.42%, respectively [16]. SC-CO_2_ treatment is shown to be an effective technology for defatting soybean powder. However, the crude protein content in soybean powder rose, from its level in CSF (38.66% ± 0.66%) to that in DSF (48.93% ± 0.82%), by approximately 26.56% after SC-CO_2_ treatment. An increase in protein content in defatted pumpkin seeds has been previously reported as a result of the relative increase in other components following fat removal [26]. It has been reported that the surface of the soybean oil body is centered around triacylglycerols, surrounded by a phospholipid layer, facilitating the creation of an O/W emulsion system without any emulsifier or high-pressure homogenization [27]. The high crude protein and low crude fat content of DSF are anticipated to have a positive impact on emulsification and stability during mayonnaise production.

### 3.2. Color Value

The findings of the color measurement of mayonnaise are displayed in Table 2. The L* values, which indicate brightness, were measured for mayonnaises containing CSF (73.2–76.55), DSF (79.43–81.96), and EY (77.53–83.22), with DSF mayonnaise showing L* values close to those of EY mayonnaise. The L* values of the emulsion could be impacted by the amount of emulsifier supplied, and it has been observed that, as the size of oil droplets reduces, light scattering rises, causing the color of the emulsion to change from gray to bright white [28]. This suggests that the concentration of emulsifiers in mayonnaise affects the L* value. As the amount of each emulsifier (in CSF, DSF, EY mayonnaise) increased, the a* and b* values tended to rise (*p* < 0.05). Mayonnaise prepared with the same amounts of emulsifiers (in CSF, DSF, and EY samples) exhibited significantly higher a* and b* values of CSF mayonnaise (*p* < 0.05). These colorimetric findings reveal that the natural color of the emulsifier affects the color of the mayonnaise [29].

The pH of mayonnaise is a crucial parameter, one that influences its shelf life, viscosity, and elasticity by reducing microbial risks [30]. Generally, the pH of mayonnaise is thought to be in the range of 3.0–4.5 [31]. The pH measurement results for mayonnaise are shown in Appendix A. CSF mayonnaise (3.82–4.34), DSF mayonnaise (3.61–4.16), and EY mayonnaise (4.17–4.67) each exhibited an increasing trend in pH values as the amount of emulsifier increased (*p* < 0.05). Mayonnaise made with DSF showed a lower pH than that made with CSF. The lower pH of DSF mayonnaise compared with CSF mayonnaise suggests potential improvements in the safety and stability of that mayonnaise.

### 3.3. Microstructures of Mayonnaise

Mayonnaise is an O/W emulsion, typically comprising 60–70% oil by total volume. The green fluorescence on the surface of the dispersed droplets indicates protein adsorption, and the red fluorescence in the discontinuous phase suggests the creation of an O/W emulsion [32]. The microstructure of mayonnaise is determined by various parameters, such as the type and amount of emulsifier used, as well as the manufacturing process, which affect factors such as the size of fat particles and the moisture in the microstructure [33]. Microstructural observations of mayonnaise are displayed in Figure 1. Both EY and plant-based mayonnaise exhibited finely dispersed spherical oil droplets in the aqueous medium. These findings suggest that soybean powder could function as an active filler in emulsion formation, surrounding the oil droplets and creating a gel network. The small droplet size in mayonnaise implied higher stability of the emulsifier [8]. Mayonnaise prepared with CSF and DSF displayed comparatively larger oil droplets than EY mayonnaise. However, at the same concentration of emulsifier, mayonnaise made with DSF showed oil droplet sizes closer to those of EY mayonnaise, and a trend was observed in which the diameter of the oil droplets decreased and became more uniform as the amount of emulsifier increased. The increase in the emulsifier stabilized the oil–water interface area and generated smaller droplets.

Changes in the droplet size observed through CLSM were consistent with the droplet size distribution and particle size results. The droplet size distribution and d_4,3_ of the CSF, DSF, and EY mayonnaises are shown in Figure 1. All of the prepared mayonnaises showed a monomodal distribution. The droplet size distribution for all mayonnaises prepared with CSF, DSF, and EY showed a trend of decreasing peak height and widening peak width as the concentration of the emulsifier rose. This result is similar to those of studies in which an increase in the amount of insoluble soy peptide aggregates led to lower peak heights and wider peak widths in droplet size distribution [34]. The particle size of mayonnaise samples containing CSF (33.01–53.35), DSF (12.02–35.84), and EY (8.3–28.98) considerably decreased as the concentration of emulsifier increased (*p* < 0.05) (Appendix A). The particle size of DSF mayonnaise showed droplet sizes closer to the particle size value of EY mayonnaise than that of CSF mayonnaise. With the same emulsifier concentration, DSF mayonnaise had smaller particle size values than CSF mayonnaise, indicating that DSF effectively adsorbed at the oil–water interface, aiding in the creation of stable droplets. DSF is anticipated to be effective in forming soy protein gel networks and is expected to be used as a substitute for EY in the production of plant-based mayonnaise [35].

### 3.4. Emulsion Capacity and Stability

The emulsion capability and stability of plant-based mayonnaise prepared with CSF and DSF and egg-based mayonnaise prepared with EY are demonstrated in Figure 2. Emulsion capacity refers to the phase separation that is due to the density difference between the dispersed and continuous phases, which in turn causes an increase in oil globules [36]. Maintaining the quality of emulsified foods like mayonnaise necessitates a high emulsion capacity, which is impacted by variables including applied pressure, vibrations, temperature variations, and the kind and concentration of emulsifiers [11]. The results of the emulsion capacity tests show that mayonnaise made with DSF and EY had an emulsion capacity of 100%, indicating stable emulsification, whereas mayonnaise made with CSF showed a capacity of about 95.7–98.3%. Similar findings with no phase separation have been reported in mayonnaise studies using chickpea protein, faba bean protein, and lentil protein emulsifiers [37]. In addition, a similar increase in emulsion stability has been observed in mustard, soy, and flaxseed emulsions after defatting, which can be attributed to the high oil absorption capacity of the defatted meal [16]. Emulsifiers, such as proteins and polysaccharides, increase viscosity by creating strong and adequate intermolecular interactions between oil droplets, thereby reducing the movement of oil droplets and eventually improving stability [38]. Thus, the increased protein content in DSF appears to enhance the oil absorption capacity of DSF and consequently stabilize the mayonnaise.

In O/W emulsion systems such as mayonnaise, temperature increases can disrupt the structure, resulting in oil exudation, thereby lowering emulsion stability [39]. The emulsion stability of mayonnaise made with CSF was 93.1–96.8% that of DSF mayonnaise, which was 97.75–99.51%, and the emulsion stability of EY mayonnaise was 97.08–98.3%, showing a decrease in emulsion stability compared with their emulsion capacity. An increasing trend in emulsion stability was observed with increased content of emulsifiers, indicating that the size of the fat globules in the mayonnaise directly impacted its stability. Larger oil droplet diameter increases the velocity of the droplets and leads to their aggregation, thus causing defects like phase separation and sedimentation [40]. Increased addition of emulsifier appears to promote protein adsorption on oil droplets, efficiently preventing emulsion coagulation. Among the three types, mayonnaise with DSF showed the highest emulsion stability. DSF, which has a larger protein concentration than CSF, appears to exhibit higher hydrophobicity, improving adsorption capacity at the droplet surface and thereby boosting emulsion capability. Higher emulsion capacity and stability have been reported with increased plant protein substitution in other plant-based mayonnaise when compared with that of EY mayonnaise [12,40].

### 3.5. Texture and Viscosity

The texture parameters of the mayonnaise are presented in Table 3. Hardness represents the resistance of a material to compression, similar to the force needed to compress food between the molars. Gumminess is generally related to the product’s elasticity or “springiness” and reflects the characteristics during chewing and swallowing [41]. The cohesiveness of mayonnaise, which indicates the internal binding strength of a food matrix and the degree to which it can deform before breaking, was found to be independent of the content of the emulsifier. Similar findings have been reported, in which the oil-to-bean ratio in mayonnaise production is unrelated to cohesiveness [42]. As the amount of emulsifier in all of the mayonnaises increased, the hardness and gumminess significantly increased, with the mayonnaise made with CSF exhibiting the lowest values at the same emulsifier concentration. At 0.5% and 1% emulsifier addition, the hardness and gumminess of EY mayonnaise were higher than those of DSF, but at 2% and 3%, DSF mayonnaise showed higher values than EY mayonnaise. This trend is comparable to the viscosity measurements (Figure 3). The viscosity results show that, after adding 1.5% emulsifier, DSF mayonnaise exhibited the highest values. Earlier studies have reported that changes in the content of protein, emulsifier, or oil in mayonnaise can influence its hardness, and that the viscosity of mayonnaise can partially impact the degree of hardness and gumminess [5]. The increased viscosity of DSF mayonnaise, with a higher total protein content than CSF mayonnaise, appears to influence its hardness and gumminess. By their nature, emulsifiers can increase the stability of O/W emulsion systems by improving the interactions within the structure and decreasing fluidity [43]. When > 2% DSF emulsifier was added to the mayonnaise, the viscosity increased and formed a stable O/W emulsion, with cohesiveness being reduced to that of EY mayonnaise. Thus, DSF might function as an efficient emulsifier through potent electrostatic interactions.

### 3.6. Sensory Evaluation

For ready-to-eat products such as mayonnaise, sensory acceptability is one of the most crucial factors. The most significant sensory characteristic of food emulsions is overall appearance, which is impacted by the combined effects of texture and color [44]. The sensory evaluation results for mayonnaise samples prepared with different emulsifiers and their concentrations are shown in Figure 4. The appearance preference for plant-based mayonnaises prepared with CSF and DSF showed a decreasing trend with an increase in the amount of emulsifier, whereas the aroma preference increased. In terms of taste and texture preference, mayonnaise with 2% DSF concentration exhibited the highest values, whereas the addition of 3% DSF negatively affected the taste and texture of the mayonnaise. In terms of overall preference, mayonnaises produced with DSF and EY were rated higher than that made with CSF, with both receiving higher scores at 2% emulsifier addition. Therefore, the potential of DSF as an EY substitute emulsifier in mayonnaise appears promising.

## 4. Conclusions

This study assesses the capacity of DSF to improve the quality of plant-based mayonnaise and replace animal-derived emulsifiers such as EY. Supercritical carbon dioxide treatment significantly reduced fat while increasing protein content in soybean powder. DSF mayonnaise exhibited fat particle characteristics similar to EY mayonnaise and demonstrated smaller, more homogeneous fat particles than mayonnaise with high CSF content, suggesting higher emulsification capability and stability. Furthermore, DSF mayonnaise demonstrated higher firmness and viscosity values, suggesting its potential as an emulsifier in various food products. In sensory assessments, mayonnaise, particularly with 2% DSF and EY concentrations, was highly rated, with 2% DSF mayonnaise receiving the highest overall preference. This study demonstrates the viability of DSF as a plant-based emulsifier in vegan mayonnaise, contributing to food development that aligns with health, environment, and vegan market demands. It also offers a foundation for sustainable, plant-based food product development without sacrificing flavor or texture. Long-term storage stability experiments should be performed in the future to further clarify the commercial applicability of DSF.

## Figures and Tables

**Figure 1 foods-13-01170-f001:**
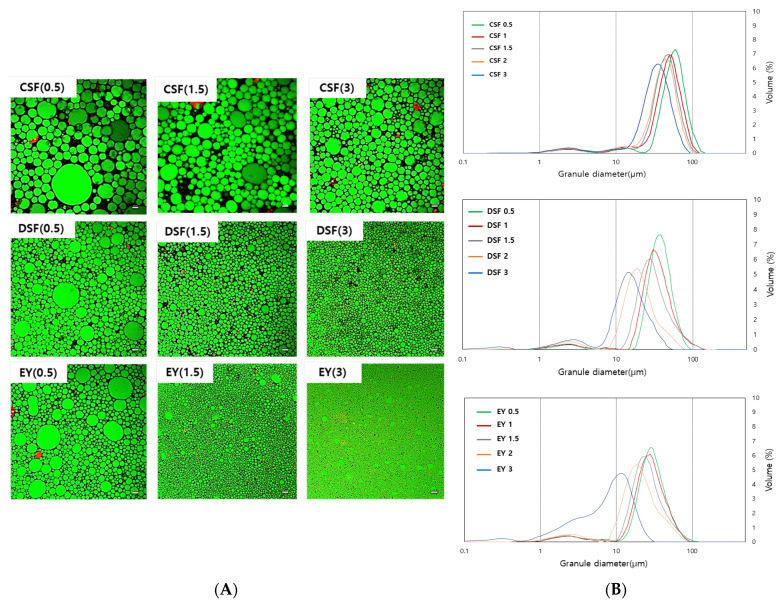
(**A**) Confocal laser scanning microscopy (CLSM) and (**B**) particle size distribution of mayonnaises prepared with control soy flour (CSF), defatted soy flour treated with SC-CO_2_ (DSF), and egg yolk (EY) at levels of 0.5, 1, 1.5, 2, and 3%.

**Figure 2 foods-13-01170-f002:**
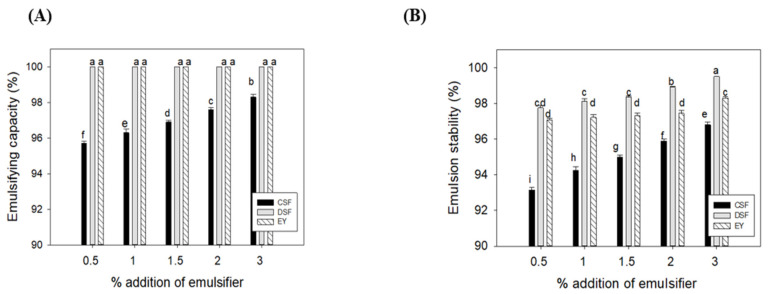
(**A**) Emulsifying capacity and (**B**) emulsion stability of mayonnaises prepared with control soy flour (CSF), defatted soy flour (DSF), and egg yolk (EY). All values are mean ± SD (*n* = 3). Different letters in bars express significant differences (*p* < 0.05).

**Figure 3 foods-13-01170-f003:**
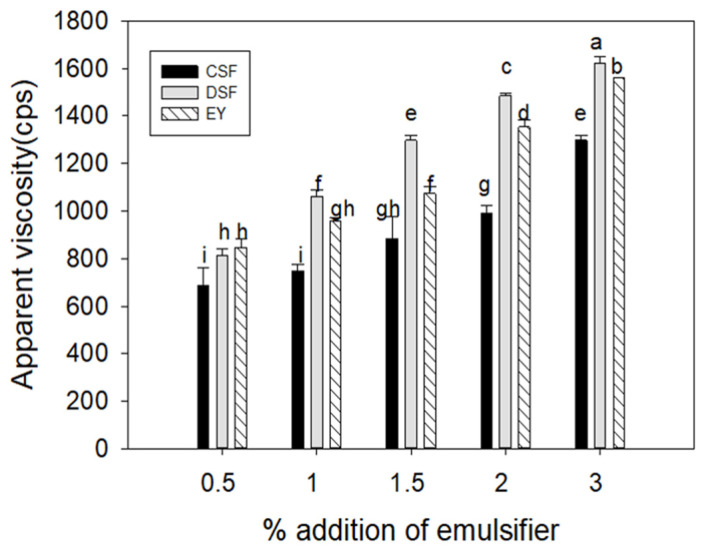
Apparent viscosity measurement of mayonnaise prepared with control soy flour (CSF), defatted soy flour (DSF), and egg yolk (EY). All values are mean ± SD (*n* = 3). Different letters in bars express significant differences (*p* < 0.05).

**Figure 4 foods-13-01170-f004:**
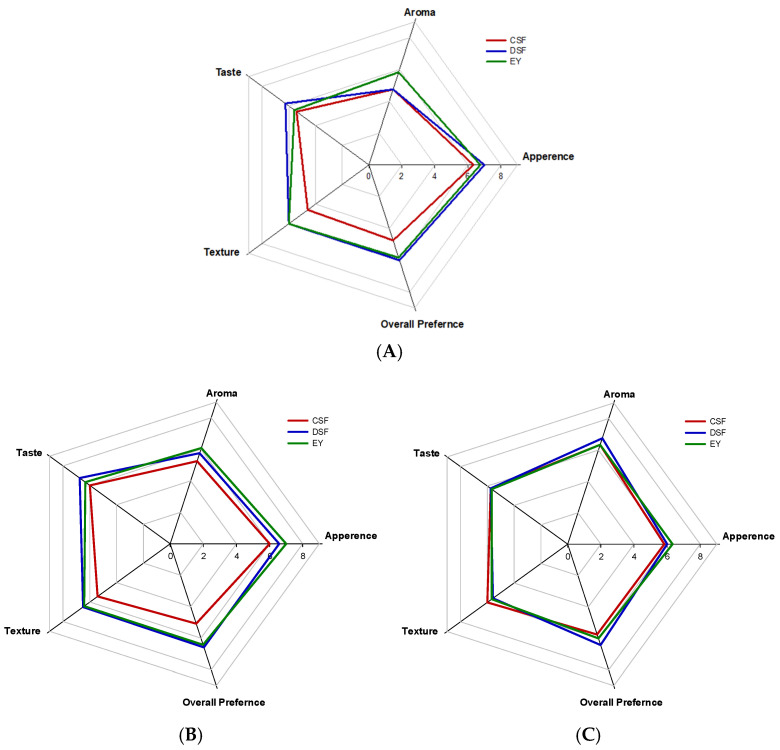
Sensory evaluation of mayonnaise prepared with control soy flour (CSF), defatted soy flour treated with supercritical carbon dioxide (DSF), and egg yolk (EY): (**A**) with 1% emulsifier, (**B**) with 2% emulsifier, (**C**) with 3% emulsifier. Rating scale: 1 (bad) ↔ 9 (good). Values are mean ± SD (*n* = 20).

**Table 1 foods-13-01170-t001:** Recipes of mayonnaise (wt. %) formulated with different types and levels of emulsifiers.

Ingredients	Emulsifier Solid Content (%)														
	CSF 0.5	CSF 1	CSF 1.5	CSF 2	CSF 3	DSF 0.5	DSF 1	DSF 1.5	DSF 2	DSF 3	EY 0.5	EY 1	EY 1.5	EY 2	EY 3
CSF	0.5	1	1.5	2	3	-	-	-	-	-	-	-	-	-	-
DSF	-	-	-	-	-	0.5	1	1.5	2	3	-	-	-	-	-
EY	-	-	-	-	-	-	-	-	-	-	0.5	1	1.5	2	3
Soybean oil	70	70	70	70	70	70	70	70	70	70	70	70	70	70	70
Sugar	2	2	2	2	2	2	2	2	2	2	2	2	2	2	2
Salt	1.5	1.5	1.5	1.5	1.5	1.5	1.5	1.5	1.5	1.5	1.5	1.5	1.5	1.5	1.5
Mustard powder	0.3	0.3	0.3	0.3	0.3	0.3	0.3	0.3	0.3	0.3	0.3	0.3	0.3	0.3	0.3
Vinegar	4	4	4	4	4	4	4	4	4	4	4	4	4	4	4
Water	22.2	22.2	22.2	22.2	22.2	22.2	22.2	22.2	22.2	22.2	22.2	22.2	22.2	22.2	22.2

Emulsifier solid content levels (five levels were used 0.5, 1, 1.5, 2, 3); CSF: control soy flour; DSF: defatted soy flour treated with SC-CO_2_; EY: egg yolk.

**Table 2 foods-13-01170-t002:** Color value of mayonnaise prepared with control soy flour (CSF), defatted soy flour (DSF), and egg yolk (EY).

Color Value		% Addition of Emulsifier				
		0.5	1	1.5	2	3
L*	CSF	73.20 ± 0.03 ^aA^	75.58 ± 0.04 ^aB^	76.55 ± 0.02 ^aD^	76.19 ± 0.03 ^aCD^	76.12 ± 0.02 ^aC^
	DSF	79.32 ± 0.01 ^aA^	79.66 ± 0.01 ^bB^	80.77 ± 0.01 ^bD^	80.62 ± 0.01 ^bC^	81.96 ± 0.00 ^bE^
	EY	77.53 ± 0.01 ^bA^	82.87 ± 0.02 ^cB^	83.22 ± 0.01 ^cD^	82.92 ± 0.02 ^cBC^	82.96 ± 0.01 ^cC^
a*	CSF	−0.64 ± 0.0 ^cA^	−0.27 ± 0.01 ^cB^	−0.07 ± 0.01 ^bC^	0.11 ± 0.02 ^cD^	0.3 ± 0.03 ^bE^
	DSF	−0.90 ± 0.01 ^bA^	−0.87 ± 0.02 ^bAB^	−0.80 ± 0.03 ^abB^	−0.76 ± 0.02 ^bC^	−0.72 ± 0.01 ^aD^
	EY	−1.05 ± 0.01 ^aA^	−0.97 ± 0.02 ^aB^	−0.84 ± 0.01 ^aC^	−0.80 ± 0.01 ^aD^	−0.77 ± 0.02 ^abDE^
b*	CSF	9.92 ± 0.01 ^cA^	11.02 ± 0.01 ^bB^	11.93 ± 0.02 ^cC^	12.68 ± 0.01 ^cD^	13.23 ± 0.02 ^cE^
	DSF	9.42 ± 0.01 ^bA^	9.59 ± 0.02 ^abB^	9.65 ± 0.01 ^bC^	9.79 ± 0.01 ^aD^	9.90 ± 0.02 ^aE^
	EY	9.30 ± 0.03 ^aA^	9.40 ± 0.00 ^aB^	9.57 ± 0.01 ^aC^	10.08 ± 0.01 ^bD^	10.98 ± 0.01 ^bE^

All values are mean ± SD (*n* = 5). Means with different letters (a–c, for the samples at different emulsifiers within the same column. A–E for the samples at different additives levels within the same row) indicate significant differences (*p* < 0.05) by Duncan’s test.

**Table 3 foods-13-01170-t003:** Texture analysis of mayonnaise prepared with control soy flour (CSF), defatted soy flour (DSF), and egg yolk (EY).

Textural Properties		% Addition of Emulsifiers				
		0.5	1	1.5	2	3
Hardness (N)	CSF	12.20 ± 0.10 ^aA^	13.93 ± 1.12 ^aB^	20.27 ± 1.31 ^aC^	25.07 ± 1.29 ^aD^	38.80 ± 4.39 ^aE^
	DSF	19.47 ± 0.84 ^bA^	22.67 ± 0.61 ^bB^	28.13 ± 1.30 ^bC^	39.37 ± 0.51 ^cD^	53.57 ± 0.15 ^bE^
	EY	22.97 ± 0.64 ^cA^	26.03 ± 0.59 ^cB^	28.73 ± 1.21 ^bC^	29.93 ± 1.83 ^bD^	39.10 ± 1.15 ^aE^
Cohesiveness	CSF	0.83 ± 0.01 ^b^	0.87 ± 0.01 ^c^	0.78 ± 0.02 ^b^	0.83 ± 0.03 ^b^	0.89 ± 0.02 ^c^
	DSF	0.82 ± 0.01 ^b^	0.81 ± 0.01 ^b^	0.82 ± 0.02 ^c^	0.79 ± 0.01 ^b^	0.80 ± 0.03 ^b^
	EY	0.75 ± 0.01 ^a^	0.72 ± 0.03 ^a^	0.72 ± 0.03 ^a^	0.76 ± 0.02 ^a^	0.75 ± 0.02 ^a^
Gumminess	CSF	10.09 ± 0.24 ^aA^	12.03 ± 1.01 ^aB^	15.80 ± 1.32 ^aC^	20.64 ± 0.84 ^aD^	34.48 ± 4.11 ^bE^
	DSF	16.06 ± 0.91 ^bA^	18.50 ± 0.54 ^bB^	23.08 ± 1.69 ^cC^	31.11 ± 1.06 ^cD^	42.83 ± 2.34 ^cE^
	EY	16.97 ± 0.62 ^bcA^	19.72 ± 1.62 ^bcB^	21.17 ± 1.64 ^bBC^	23.90 ± 1.59 ^bC^	29.49 ± 1.03 ^aD^

All values are mean ± SD (n = 5). Means with different letters (a–c, for the samples at different emulsifiers within the same column. A–E for the samples at different additives levels within the same row) indicate significant differences (*p* < 0.05) by Duncan’s test.

## Data Availability

The original contributions presented in the study are included in the article/Appendix A, further inquiries can be directed to the corresponding author.

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
