# Peer review of "Quality Characteristics of Vegan Mayonnaise Produced Using Supercritical Carbon Dioxide-Processed Defatted Soybean Flour"

_foods, 2024, doi:10.3390/foods13081170_

Round 1

Reviewer 1 Report

Comments and Suggestions for Authors

The manuscript from Han et al.  propose defatted soybean powder as replacement for animal ingredient in mayonnaise. For this purpose, the authors studied the potential of defatted soybean powder, obtained by supercritical carbon dioxide extraction, in the improvement of the quality of plant-based mayonnaise. The study involved the production of the defatted soybean powder (DSF-SC), assess of its main quality attributes, which were compared with those of mayonnaise made with control soy flour (CSF), as well as with that produced with egg yolk (EY). The authors found that the use of DSF-SC, with increased emulsifier quantity, lead to better emulsion stability, viscosity, together with smaller and more uniform particle size, compared to the control CSF mayonnaise. Moreover, sensory evaluation of the produced mayonnaise was also carried out and DSF-SC mayonnaise was rated higher in sensory evaluation. To conclude the authors found that DSF-SC can be a promising plant-based alternative emulsifier for replacing animal ingredient, with the present study showing that that  up to 2% addition of DSF-SC improves the emulsion and sensory properties of the product.

The subject is state of the art, since consumer demanding for more natural and green products is in line with the use of plant based ingredients for the formulation of processed products. Additionally, the use of a eco-friendly technique to obtain the defatted soybean powder (i.e. supercritical CO2 extraction - considerer a green suitable alternative to hazards organic solvent extraction) is also a bonus to the “greenness” of the final product. In this regard, the present study addresses topics of interest.  On the other hand, the research methodology is sound and all the evaluated attributes are in line with the experimental protocols required for these type of studies. Overall, this is in general a very good paper. The study presents interesting and new results and contribute to the literature. I recommend the publication of the manuscript with some minor revision of the English. Moreover, a few minor correction concerning the scientific content of the paper should also be carried out. Some comments and suggestions are presented below:

- The composition of DSF-SC, CSF and EY (table S2) show a considerable difference in the crude fat content of the three ingredients. The authors should add to the text some considerations regarding the effect of using a low-fat ingredient (DSF-SC) compared to a high-fat (EY) and a moderate fat (CSF). Particularly, the importance and the effect of the presence (or its absence) of lipid fraction in the ingredients in the emulsion

- The moisture content of the three tested ingredients (table S2) showed that control SF presented twice moisture content than DSF and EY. Taking into account that mayonnaise recipe used the same amount of vinegar and water for all tested ingredients, this could have an effect in the microstructure results obtained for each prepared maionese. Particularly for the tests with higher emulsifier concentration. The authors could comment this possibility and add some comments on the manuscript regarding this results.

Author Response

Dear Editor,

We would like to thank you for your consideration and suggestions on our manuscript “Quality characteristics of vegan mayonnaise produced using supercritical carbon dioxide-processed defatted soybean flour”. We have found the comments helpful and our revised manuscript represents a significant improvement over our initial submission. We have reviewed the comments of the reviewers and have accordingly revised the manuscript. In the following pages, there are our point-to-point responses to each of the comments of the reviewers.

Reviewers’ comments

Reviewer #1

[1]

Comment: The composition of DSF-SC, CSF and EY (table S2) show a considerable difference in the crude fat content of the three ingredients. The authors should add to the text some considerations regarding the effect of using a low-fat ingredient (DSFSC) compared to a high-fat (EY) and a moderate fat (CSF). Particularly, the importance and the effect of the presence (or its absence) of lipid fraction in the ingredients in the emulsion
Response: As the reviewer suggests, we would like to bring to your notice that after defatted soy flour, protein content increased. In addition, its emulsification capacity was also enhanced. This property is used in our study and has proven to be a good plant-based replacement for egg yolk. The main reason for selecting supercritical CO2 extraction is due to its advantage- no protein denaturation, compared to hexane extraction, and this protein helps maintain the emulsion stability. Most of this information has been discussed in paragraphs 3 and 4 of the introduction part.

[2]

 Comment: The moisture content of the three tested ingredients (table S2) showed that control SF presented twice moisture content than DSF and EY. Taking into account that mayonnaise recipe used the same amount of vinegar and water for all tested ingredients, this could have an effect in the microstructure results obtained for each prepared maionese. Particularly for the tests with higher emulsifier concentration. The authors could comment this possibility and add some comments on the manuscript regarding this results.

Response: The moisture content of the mayonnaise was adjusted by adding the same amount solid content of the emulsifier and same amount of water to the mayonnaise. By this way, we ensured that the moisture content of the mayonnaise remained the same in all mayo samples, and did not have any influence on the microstructure of the mayonnaise samples.

Reviewer 2 Report

Comments and Suggestions for Authors

This paper aims to making defatted soybean powder treated with supercritical carbon dioxide (DSF-SC), a promising plant-based alternative emulsifier for replacing animal ingredients. The authors evaluated the color, pH, microstructure, particle size, physical and thermal stability, viscosity, texture, and sensory evaluation. It is a very interesting paper, but the expression needs to improve.

1) The authors did not add the line number.

2) In the abstract, the abbreviations are too many. Line 6, which substance is the “emulsifier”? It should be “with increased quantity of DSF-SC emulsifier”.

3) In each table, the abbreviations should be noted. In the text, the abbreviations are too many and it is not easy to read this paper.

4) Table S1 is important in this paper, but has no units. It should be inserted into the text, otherwise it is difficult to read this paper. In table S2, why is there not the content of dietary fiber? The soybean protein is easy to dissolve water, and the soluble soybean polysaccharides are now used for emulsifier, in the discussion section, the emulsion capability from control soy flour and defatted soybean powder treated with supercritical carbon dioxide should be discussed. The other question is, is there the beany flavor in the final products?

5) In table S4, what is d4,3?

6) The principal component analysis had better be used for this study.

7) In table 2, cohesiveness has not unit.

Author Response

Dear Editor,

We would like to thank you for your consideration and suggestions on our manuscript “Quality characteristics of vegan mayonnaise produced using supercritical carbon dioxide-processed defatted soybean flour”. We have found the comments helpful and our revised manuscript represents a significant improvement over our initial submission. We have reviewed the comments of the reviewers and have accordingly revised the manuscript. In the following pages, there are our point-to-point responses to each of the comments of the reviewers.

Reviewers’ comments

Reviewer #2

[1]

Comment: The authors did not add the line number.

Response: We would like to thank the reviewer for bringing this to our notice. We have inserted the line numbers in our text.

[2]

Comment: In the abstract, the abbreviations are too many. Line 6, which substance is the “emulsifier”? It should be “with increased quantity of DSF-SC emulsifier”.

Response: As the reviewer suggested, we agree with the statement, and hence we have rectified it to “It was found that mayonnaise made with an increased quantity of DSF emulsifier had better emulsion stability, viscosity, and a smaller, more uniform particle size compared to that of the control.”

 [3]

Comment: In each table, the abbreviations should be noted. In the text, the abbreviations are too many and it is not easy to read this paper.

Response: We would like to bring to the notice of the reviewers that we have used only 3 abbreviations in the abstract, i.e. CSF for Control soy flour, DSF for Defatted soy flour (we have replaced DSF-SC with DSF), and EY for egg yolk, which are the important aspects in our manuscript.

[4]

 Comment: Table S1 is important in this paper but has no units. It should be inserted into the text, otherwise it is difficult to read this paper. In table S2, why is there not the content of dietary fiber? The soybean protein is easy to dissolve water, and the soluble soybean polysaccharides are now used as emulsifier, in the discussion section, the emulsion capability from control soy flour and defatted soybean powder treated with supercritical carbon dioxide should be discussed. The other question is, is there the beany flavor in the final products?

Response: For the kind information of the reviewers, table S1 has been inserted in-text and renamed as table 1. Also, the beany flavor was well reduced in the mayonnaise. Supercritical fluid extraction is known to extract the volatile components, especially the fatty acids in soybean responsible for its beany flavor to a greater extent (Ibáñez, Mendiola, & Castro-Puyana, 2016).  The emulsion stability of the mayonnaise and the emulsion capacity of the emulsifier have been discussed in the result and discussion part from line 276.

Ref: Ibáñez, E., J. A. Mendiola, and M. Castro-Puyana. "Supercritical fluid extraction." (2016): 227-233.

[5]

 Comment: In table S4, what is d4,3?

Response: As per the suggestions made by the reviewer, for better understanding, we have changed the unit to µm.

[6]

 Comment: The principal component analysis had better be used for this study.

Response: We used statistical program of Duncan test (SAS 9.4). We have added the sentence (Table 2 and Table 3).

[7]

Comment: In table 2, cohesiveness has not unit.

Response: In our study, only hardness has been given a unit newton (N) whereas the rest of the texture parameters have no unit.

Reviewer 3 Report

Comments and Suggestions for Authors

This paper, treating defatted soybean powder with supercritical carbon dioxide to improve the quality of plant-based mayonnaise, assesses its capacity to improve the quality of plant-based mayonnaise and replace animal-derived emulsifiers. Overall, the article is rich in data and proper analysis. However, there are still some issues that need further revision.

1.  2.4. Plant-Based Mayonnaise Preparation: Except for the CSF, DSCF-CS, and EY, the formulations of the other substances are the same in the recipe for mayonnaise. However, in the preparation of mayonnaise, there are not the same processes and parameters for the preparation of plant-based mayonnaise and regular mayonnaise. What are the reasons for this difference? This may have some impact on the results of the controlled trials. It is suggested that the authors could further explain the experimental design.

2.  2.8. Measurement of Physical and Thermal Stability: Suggested to provide literature support for these measurement methods.

3.  3.3. Microstructures of Mayonnaise: Figure 1 (b) displays the particle size analysis of mayonnaise with different emulsifiers at different concentrations. However, in Figure (b) B (De-Fatted Soy Flour), at 2% and 3% content, another region of wide distribution appeared to the right of the peak of the single peak and with a larger particle size, and it didnt occur in A and C. Is this due to the inhomogeneous distribution and unstable of the mayonnaise with DSF-SC?  The author didnt explain this phenomenon.

4. 3.4. Emulsion Capacity and Stability: There's a sentence in this section that says “Increased oil droplet diameter can cause defects like phase separation and sedimentation, which can be separated by gravity”. I can't seem to understand exactly the meaning of this sentence. Is the author trying to convey that due to gravity, the increased oil droplet diameter tends to lead to phase separation and sedimentation? Please the author describes more accurately. 

53.4. Emulsion Capacity and Stability: What is the author trying to illustrate by citing the examples of both clover protein mayonnaise and mayonnaise containing wheat germ protein? It is suggested that the author rationalize the logic and change the expression.

6The format in Table 2 suggests adjustments.

7. It is suggested to add the data and results of the highlights of this experiment in the conclusion section to increase the persuasiveness of the conclusion. And please also make recommendations for future research, such as what more needs to be done.

8. It is very interesting that make vegan mayonnaise using soybean powder. However, the naturally occurring beany flavor in soybeans restricts their further development. Is there a beany flavor to plant-based mayonnaise?

Author Response

Dear Editor,

We would like to thank you for your consideration and suggestions on our manuscript “Quality characteristics of vegan mayonnaise produced using supercritical carbon dioxide-processed defatted soybean flour”. We have found the comments helpful and our revised manuscript represents a significant improvement over our initial submission. We have reviewed the comments of the reviewers and have accordingly revised the manuscript. In the following pages, there are our point-to-point responses to each of the comments of the reviewers.

Reviewer #3

[1]

Comment: 2.4. Plant-Based Mayonnaise Preparation: Except for the CSF, DSCF-CS, and EY, the formulations of the other substances are the same in the recipe for mayonnaise. However, in the preparation of mayonnaise, there are not the same processes and parameters for the preparation of plant-based mayonnaise and regular mayonnaise. What are the reasons for this difference? This may have some impact on the results of the controlled trials. It is suggested that the authors could further explain the experimental design.

Response: We made a mistake in describing the processing steps followed for the preparation of mayonnaise samples. There was no heat involvement in the processing. It has been rectified in the manuscript (lines 97-108).

[2]

 Comment: 2.8. Measurement of Physical and Thermal Stability: Suggested to provide literature support for these measurement methods.

Response: The measurement of physical and thermal stability was carried out by following the method of Lee et al., 2022 with slight modifications.

[3]

Comment: 3.3. Microstructures of Mayonnaise: Figure 1 (b) displays the particle size analysis of mayonnaise with different emulsifiers at different concentrations. However, in Figure 1 (b) B (De-Fatted Soy Flour), at 2% and 3% content, another region of wide distribution appeared to the right of the peak of the single peak and with a larger particle size, and it didn’t occur in A and C. Is this due to the inhomogeneous distribution and unstable of the mayonnaise with DSF-SC?  The author didn’t explain this phenomenon.

Response: This was a slight statistical error caused during the analysis. After reanalysing no such differences are noted. The rectified image is shown above.

[4]

Comment: 3.4. Emulsion Capacity and Stability: There's a sentence in this section that says “Increased oil droplet diameter can cause defects like phase separation and sedimentation, which can be separated by gravity”. I can't seem to understand exactly the meaning of this sentence. Is the author trying to convey that due to gravity, the increased oil droplet diameter tends to lead to phase separation and sedimentation? Please the author describes more accurately. 

Response: We have rectified the sentence as per the reviewer’s suggestion “Larger oil droplet diameter increases the velocity of droplets and causes their aggregation, and hence can cause defects such as phase separation and sedimentation.” (line no: 283-285)

[5]

Comment: 3.4. Emulsion Capacity and Stability: What is the author trying to illustrate by citing the examples of both clover protein mayonnaise and mayonnaise containing wheat germ protein? It is suggested that the author rationalize the logic and change the expression.

Response: As per the reviewer’s suggestion, the statement has been rectified to “Higher emulsion capacity and stability were reported with increased plant-protein substitution in other plant-based mayonnaise compared to that of EY mayonnaise.” (line no: 290-292)

[6]

Comment: The format in Table 2 suggests adjustments.

Response: As per the suggestions of the reviewer, we have made the adjustments in the manuscript (page no: 6).

[7]

Comment: It is suggested to add the data and results of the highlights of this experiment in the conclusion section to increase the persuasiveness of the conclusion. And please also make recommendations for future research, such as what more needs to be done.

Response: As the reviewer suggested, we have added the following sentence “It is worth noting that supercritical CO2 extraction of fats from soy flour made it a better vegan alternative to full fat soy flour in making the mayonnaise, because of better water holding capacity, and enhanced protein content. Hence, DSF mayonnaise can be considered the best alternative to traditional mayonnaise, especially for those who are interested in a vegan-friendly and healthy replacement.

[8]

Comment: It is very interesting that make vegan mayonnaise using soybean powder. However, the naturally occurring beany flavor in soybeans restricts their further development. Is there a beany flavor to plant-based mayonnaise?

Response: The beany flavor was well reduced in the mayonnaise. Supercritical fluid extraction is known to extract the volatile components, especially the fatty acids in soybean responsible for its beany flavor to a greater extent (Ibáñez, Mendiola, & Castro-Puyana, 2016).

Ref: Ibáñez, E., J. A. Mendiola, and M. Castro-Puyana. "Supercritical fluid extraction." (2016): 227-233.

Reviewer 4 Report

Comments and Suggestions for Authors

Dear authors,

Please consider the following suggestions to correct the manuscript.

Topic 2.2

Include how much of raw material was used. Include also the vessel volume. Include mass flow rate or the quantity of carbon dioxide spent per batch. What was the condition in the separator?

Topic 3.1

''Supercritical carbon.... powders'' - Please explain why. Water from sample cannot extracted, so probably the temperature used contributed to decrease the moisture.  

''The increase in... removal'' - Actually the protein was already present in matrix. The SFE of fat worked as a physical pretreatment to enhance the availability of other molecules, and consequently, increase their detection during analysis.

Topic 3.2

''This suggests... value'' - Did the authors measure the degree of emulsification or are you referring to the concentration of emulsifier (table 1)? Explain how color coordinates measured may be affected with that.

Topic 3.6

''In terms...promising'' - Interesting! I recommend include a sentence on how these results will support further studies (consumer's preference with the optimal formulation). 

Author Response

Reviewer #4

[1]

Comment: Topic 2.2 Include how much of raw material was used. Include also the vessel volume. Include mass flow rate or the quantity of carbon dioxide spent per batch. What was the condition in the separator?

Response: Kindly note that 500g of soy flour was used for defattation, and the flow rate of the CO2 was 30g/min. More details about our procedure can be found in our previous paper by Kang et al., 2017 (line no: 90).

[2]

Comment: Topic 3.1 ''Supercritical carbon.... powders'' - Please explain why. Water from sample cannot extracted, so probably the temperature used contributed to decrease the moisture.  

Response: As the reviewer pointed out, the decrease in the moisture content of DSF is mainly because the solubility of water is increased in CO2 under supercritical conditions (temp 40-60ºC and pressures below 5 MPa) (Sabirzyanov et al., 2002; King et al., 1992).

''The increase in... removal'' - Actually the protein was already present in matrix. The SFE of fat worked as a physical pretreatment to enhance the availability of other molecules, and consequently, increase their detection during analysis.

Ref.1 Sabirzyanov, A. N., Il'in, A. P., Akhunov, A. R., & Gumerov, F. M. (2002). Solubility of water in supercritical carbon dioxide. High Temperature, 40(2), 203-206.

Ref.2 King, M. B., Mubarak, A., Kim, J. D., & Bott, T. R. (1992). The mutual solubilities of water with supercritical and liquid carbon dioxides. The Journal of Supercritical Fluids, 5(4), 296-302.

[3]

Comment: Topic 3.2 ''This suggests... value'' - Did the authors measure the degree of emulsification or are you referring to the concentration of emulsifier (table 1)? Explain how color coordinates measured may be affected with that.

Response: Emulsion lightness (L* value) increased with increased droplet concentration and decreased droplet size (30-0.2 µm) (Chantrapornchai et al., 1998).” As the soybean oil was added during the preparation of DSF mayonnaise, it is also possible that the addition of soybean oil to reduced-fat samples led to lightness (L*) of the sample, and this phenomenon is caused by significant light reflectance from the small fat globules in the oil (Youssef and Barbut, 2011). SC-CO2 also has it effect on color, because the low polarity of oils and pigments helps in their extraction (Khanra et al., 2018), suggesting that this technique might have extracted out certain pigments leading to whiter powder (Laurent et al., 2022).

Ref.1 Chantrapornchai, W., Clydesdale, F., & McClements, D. J. (1998). Influence of droplet size and concentration on the color of oil-in-water emulsions. Journal of Agricultural and Food Chemistry, 46(8), 2914-2920.

Ref.2 Youssef, M. K., & Barbut, S. (2011). Effects of two types of soy protein isolates, native and preheated whey protein isolates on emulsified meat batters prepared at different protein levels. Meat science, 87(1), 54-60.

Ref.3 Khanra, S., Mondal, M., Halder, G., Tiwari, O. N., Gayen, K., & Bhowmick, T. K. (2018). Downstream processing of microalgae for pigments, protein and carbohydrate in industrial application: A review. Food and bioproducts processing, 110, 60-84.

Ref.4 Laurent, S., Jury, V., de Lamballerie, M., & Fayolle, F. (2022). Effect of two defatting processes on the physicochemical and flow properties of Hermetia illucens and Tenebrio molitor larvae powders. Journal of Food Processing and Preservation, 46(10), e16853.

[4]

Comment: Topic 3.6 ''In terms...promising'' - Interesting! I recommend include a sentence on how these results will support further studies (consumer's preference with the optimal formulation). 

Response: It is interesting to note that there are very limited studies related to the application of defatted soy flour in the preparation of mayonnaise. It is worth noting that defattation of soy flour using supercritical CO2 extraction made it a better vegan alternative to full fat soy flour in making the mayonnaise, because of better water-holding capacity, water absorption capacity, oil absorption capacity, emulsion stability, emulsion capacity, and enhanced protein content

Reviewer 5 Report

Comments and Suggestions for Authors

This study describes the potential of defatted soybean powder treated with supercritical carbon dioxide in improving the quality of vegan mayonnaise. The paper is written quite well but I noticed the following:

Table S1 Mayonnaise recipes varying emulsifiers and additive amounts, is important for further understanding of the text, it should be presented in the material and methods section.

Describe on the basis of which the variations in the amount of added emulsifier were chosen

Equation 2 and 3, remove, the number as a label is enough. The equations are the same?

Comment in more detail on the content of ash, i.e. minerals, in the three tested types of mayonnaise.

My proposal is to include the tables from the supplementary material in the text for easier monitoring and understanding of the results and discussion.

Comment on the influence of the addition of different concentrations of emulsifiers on the color and emulsion capacity of mayonnaise.

Why statistical analysis was not applied that would identify optimal mayonnaise sample formulation from the aspect of investigated quality parameters?

Add a short discussion on the applicability and economics of producing this type of mayonnaise in the industry.

Author Response

Reviewer #5

[1]

Comment : Table S1 Mayonnaise recipes varying emulsifiers and additive amounts, is important for further understanding of the text, it should be presented in the material and methods section.

Response: As per reviewer suggestion, we have added the mayonnaise recipe as table 1 in the material and methods section (Page:3)

[2]

Comment : Describe on the basis of which the variations in the amount of added emulsifier were chosen

Response: From our initial trials, we have noticed that an emulsifier over a certain level (>2%) fails to give the proper emulsification and functional properties to the mayonnaise. Therefore, we set up this range of emulsifier addition in our study.

[3]

Comment: Equation 2 and 3, remove, the number as a label is enough. The equations are the same?

Response: As the reviewer suggested, we have rectified it by removing the label of the equation.

[4]

Comment : in more detail on the content of ash, i.e. minerals, in the three tested types of mayonnaise.

Response: Since our study mainly focused on the development of defatted soy flour by supercritical CO2 extraction and using it for the preparation of mayonnaise, much of the experiments carried out mostly emphasized analysing its fat and protein content, emulsifying capacity, and textural studies.

[5]

Comment: My proposal is to include the tables from the supplementary material in the text for easier monitoring and understanding of the results and discussion.

Response: As the journal has certain restrictions on the number of intext figures and tables, we will be adding only Table S1 (preparation of mayonnaise) into the main text. Page no:3

[6]

Comment : on the influence of the addition of different concentrations of emulsifiers on the color and emulsion capacity of mayonnaise.

Response: As the reviewer has pointed out, the effect of different concentrations of emulsifiers on color and emulsion has been discussed by the author from lines 200-205 of the manuscript. It has been noticed that with an increase in emulsifier levels, the emulsion capacity also increased. Since protein plays an important role in emulsification, it might be the enhanced protein content in the soy flour that plays the role of enhancing the emulsifying capacity, especially after defattation with SC-CO2 as observed in our previous study (Kang et al., 2017).

Ref.1 Kang, S.W.,  Rahman, M. S.,  Kim, A.N.,  Lee, K.Y.,  Park, C.Y.,  Kerr, W. L., Choi, S.G. (2017). Comparative study of the quality characteristics of defatted soy flour treated by supercritical carbon dioxide and organic solvent. Journal of Food Science and Technology. 54, 2485-2493.

[7]

Comment: Add a short discussion on the applicability and economics of producing this type of mayonnaise in the industry.

Response: As our study mainly focused on reducing the fat content of soy flour using supercritical CO2 extraction and utilizing it in the preparation of mayonnaise, much of the focus was done on studying its physicochemical characteristics rather than on an economic point of view.

Round 2

Reviewer 3 Report

Comments and Suggestions for Authors

In this paper, treating defatted soybean powder with supercritical carbon dioxide to improve the quality of plant-based mayonnaise, assesses the potential for quality enhancement of vegan mayonnaise using supercritical treatment. Overall, the data of this paper is relatively substantial and the analysis is proper. The authors have made sufficient modifications according to the modification comments. The manuscript in its present version is apposite for publication in Foods, and I suggest that this paper be accepted without further modification.